# The Improved Phytoextraction of Heavy Metals and the Growth of *Trifolium repens* L.: The Role of K₂HEDP and Plant Growth Regulators Alone and in Combination

**Anna Makarova** [1,*] , **Elena Nikulina** [2], **Tatiana Avdeenkova** [1] **and Ksenia Pishaeva** [1]

1. Green Chemistry for Sustainable Development, Mendeleev University of Chemical Technology of Russia, 125047 Moscow, Russia; avdeenkovats@mail.ru (T.A.); hurts.ivanova@yandex.ru (K.P.)
2. Non-Profit Institute for Chemical Reagents and High Purity Chemical Substances of NRC «Kurchatov Institute», 107076 Moscow, Russia; nikulina_elena@mail.ru
* Correspondence: annmakarova@mail.ru

**Abstract:** Heavy metals are among the most widespread pollutants in soil. Phytoextraction technology is used to solve the problem of multi-metal-contaminated soil. The efficiency of this process can be increased by introducing various amendments. A soil amendment is any material added to a soil to improve its physical properties, such as water retention, permeability, water infiltration, drainage, aeration, and structure. Some chemical amendments for enhanced phytoextraction, such as amino polycarboxylates chelators, can be hazardous to the environment and perform poorly at pH > 8. The effect of the potassium salt of hydroxyethylidene diphosphonic acid (K₂HEDP), plant growth regulators (PGRs), and iron chelate alone and in combination on the phytoextraction by *Trifolium repens* L. seedlings of Cd, Ni, and Cu was studied in this work. K₂HEDP works in a wider pH range. The results of this study confirmed that amino polycarboxylate chelators, with the sodium salt of ethylene diamine tetraacetic acid (Na₂EDTA) as an example, have a pronounced negative effect on the growth and development (organ mass) of *Trifolium repens* L. seedlings. K₂HEDP, proposed by the authors instead of Na₂EDTA, produced a pronounced positive effect on plant growth and development, which was further enhanced by the use of PGRs and with iron chelates. However, it should be noted that K₂HEDP showed significantly lower efficiency in trials on the *Trifolium repens* L. seedlings. The highest was the efficiency of K₂HEDP with PGRs and iron chelates for the phytoextraction of Cd.

**Keywords:** multi-metal-contaminated soil; phytoextraction; chelating ligand; plant growth regulators; complexones—organophosphorus; iron chelate

## 1. Introduction

The intensive developments of industrial production and the scientific–technological revolution have led to a high level of chemical pollution [1]. The most significant sources of heavy metal pollution are high-temperature metallurgical industries [2], thermal power plants [3], the mining industry [4], road transport [5], and municipal solid waste (MSW) landfill sites [6].

Heavy metals are among the most widespread pollutants of the environment and living organisms [7] and are among the priority pollutants [8]. Soil pollution with heavy metals is very persistent and complex [9]. Metals accumulated in the soil cover undergo practically no biodegradation. They are very slowly removed during leaching, water, and wind erosion [10]. Most often, a significant excess of the established standards is observed for many metals simultaneously in contaminated native soils [11]. The risk caused by soil contamination with heavy metals is all the greater because they can accumulate in small doses over a long period of time [12], and contamination manifests itself after their content reaches the maximum permissible values.

Being in high concentrations in the soil horizon, heavy metals can accumulate in plants [13], serving as the basis for the development of phytoextraction technology, such

as cleaning contaminated soils using phytoextractor plants [14]. As noted in the work by Barbafieri et al. [11], this technology is receiving more and more attention as it has zero impact on the environment. Early studies [15] found that natural phytoextraction is a slow-moving process due to the inability to simultaneously remove significant amounts of metals from the soil using plants under normal agronomic conditions [5].

At the present stage, an increase in the efficiency of phytoextraction is strictly associated with such key factors as:

- an increase in the solubility and bioavailability of metal ions in the soil substrate;
- an increase in the potential of plants to accumulate pollutants in terrestrial organs and to develop large biomass;
- the physiological tolerance of plants to stress, caused by high concentrations of pollutants.

In this vein, the strategy of assisted phytoextraction [16] using chemical reagents and/or other amendments, individually or in combination, that interact with heavy metal cations has become widespread and contributes to multiple increases in the absorption of the latter by plants on the one hand, and, on the other hand, to increase the viable threshold of the accumulating plants themselves.

Currently, one of the most studied and developed methods is chemical reagent-assisted phytoextraction [17], which has been the subject of numerous studies [18,19]. Within the framework of this technology, the influence of various chelating agents, primarily synthetic polyaminopolycarboxylic acids such as ethylene diamine tetraacetic acid (EDTA) [20], nitrilotriacetic acid (NTA) [21], and ethylenediamine-*N*,*N*′-disuccinic acid (EDDS) [22], on the efficiency of the absorption of heavy metal ions has been well-studied. Despite the high versatility of the latter and the ability to form water-soluble complexes with almost all heavy metals, their range of action is limited by high pH values of the medium (pH > 8) [23]. Unfortunately, most synthetic polyaminopolycarboxylic acids are not readily biodegradable [24]. EDTA also dissolves heavy metals from sediments and soils [25], increasing their mobility, which can enhance their toxicity in water bodies [26].

Another well-known approach to stimulating phytoextraction is to apply phytohormonal amendments (plant growth regulators (PGRs)), the action of which is mainly aimed at increasing the biomass and improving the stress resistance of plants. Herein, our focus is on PGR-assisted phytoextraction. As an example of phytohormones and PGRs, Hadi et al. [27] reported an improvement in the phytoextraction of Pb in maize owing to plant treatments with compounds such as GA (gibberellic acid) and IAA (indoleacetic acid) [28], as well as improvement in the phytoextraction of Pb in alfalfa. In some cases, significant uptake is observed at 2800% with the combined use of a chelating agent (EDTA) and PGRs [27].

The phytoextraction of multicontaminated soils is much more difficult than that of soils contaminated mainly with one heavy metal [11] due to the heterogeneous physicochemical properties of metals and metalloids (the ability to immobilize and precipitate, sorption, and complexation reactions) and the cumulative toxic effect on phytoextractor plants [29].

One of the first signs of plant intoxication with heavy metals is the inhibition of plant growth [30]. High concentrations of heavy metals in the medium can also suppress the intensity of photosynthesis [31], enzymatic activity [32], respiration [33] and absorption of nutrients [34] and cause secondary chlorosis [35] etc. The toxic effects of Co, Cd, Cu, and Ni are similar [36–38].

The further development and improvement of phytoextraction technology are closely related to the search for new chelators and effective combinations of various amendments that, in combination with one another, can demonstrate the best results. In the case of soil remediation with polymetallic contamination, the strategy of selecting a combination of amendments becomes especially important. This study is devoted to the selection of amendments for phytoextraction used to clean MSW landfill sites from polymetallic contaminants. In this case, the ability to work at pH > 8 is a critical parameter for phytoextraction. This is because the pH in MSW landfill sites, according to the authors' research,

can be more than 9. Other researchers have also confirmed the possibility of an MSW landfill site of pH > 8 [39]. It is also an important condition in the selection of amendments that these amendments do not harm the environment.

## 2. Problem Formulation

Within the framework of this study of the phytoextraction of soils with polymetallic contamination, it was of interest to solve the following problems:

1.  Testing a new potential chemical inducer of a compound from the class of bisphosphonates—synthetic phosphorus-containing complexones. Organophosphorus compounds, such as hydroxyethylidene diphosphonic acid (HEDP), are capable of forming stable water-soluble complex compounds with many heavy metals in pHs of >9. As an analogy of natural pyrophosphates, HEDP is involved in more than 60 biochemical cellular reactions by regulating ionic calcium and phosphorus exchange. Additionally, organophosphorus is considerably less toxic to living systems and organisms than carboxyl-containing complexes. There is a known study on the use of HEDP for the phytoextraction of Cd, where HEDP has shown greater efficiency [40]. However, there are no studies on the effectiveness of using HEDP for polymetallic contaminants.

2.  The application of combined treatment with various functional corrections that allows simultaneous stimulation of the absorption of heavy metal ions, photosynthesis, and biomass growth. The complex scheme should be based on a combination of treatments with a chelating agent, an iron complexonate, and hormonal supplements. Taking into account the pronounced manifestation of the antagonism of metal ions during the phytoextraction of multicontaminated soils, the authors of this work put forward the hypothesis that the correction of iron deficiency can have a positive effect on (the overall efficiency of the process) stimulation of photosynthesis and can lead to an improvement of the general physiological state of the latter. Furthermore, the additional use of PGRs can also reduce the stress caused by high concentrations of pollutants and can compensate for the negative effect of the latter on biomass growth.

## 3. Materials and Methods

This research aimed to experimentally study the use of $K_2HEDP$ (double-substituted potassium salt of HEDP) as a chelating agent individually and in combination with exogenous treatments with PGRs from the group of auxins and gibberellins for the phytoextraction of multicontaminated substrates (Cd, Ni, and Cu). Ni and Cu were chosen as the main pollutants, because the analysis of soil samples in an MSW landfill site in Moscow revealed that the levels of these heavy metals in the soil exceed their normal levels more than seven times. Moreover, some authors have determined Cd as a key factor of the negative impact of leachate on groundwater quality [41].

Cu, Ni, and Cd form complexes with HEDP, and their stability constants characterize the strengths of these chelate compounds. The effect observed from the application of $K_2HEDP$ was compared to the effect observed from the use of the $Na_2EDTA$ (double-substituted sodium salt of EDTA).

Additionally, taking into account the pronounced manifestation of the antagonism of metal ions during the phytoextraction of multi-metal-contaminated soils, the authors of this work propose that additional treatment of plants with iron chelate—Na (FeEDDHA) (sodium salt of the ethylenediamine-*N*,*N*′-bis(hydroxyphenyl) acetic acid of iron).

To fulfil the goal set by the authors, a laboratory study of the phytoextraction of heavy metals from a simulated contaminated substrate was carried out using various amendments.

The effect observed from the application of $K_2HEDP$, including a combination of exogenous treatments with PGRs and Na (FeEDDHA), was compared to the effect observed from the use of $Na_2EDTA$. Moreover, the effectiveness of the amendments on multicontam-

inated substrates (Cd, Ni, and Cu) was compared to the effect observed for mononickel contamination.

### 3.1. Materials

Standard laboratory equipment, balanced with an accuracy of $\pm 0.1$ mg, and plastic vegetation pots for planting seeds with volumes of 1 L were used.

### 3.1.1. Research Objects

*Trifolium repens* L. (white clover) seedlings were the focus of this research. Clover is a very common wild-growing crop in the territory of the Russian Federation. In total, there are more than 70 growing species of clover within the country. *Trifolium repens* L., belonging to leguminous herbaceous crops, is kept in herbage for 2–3 years. The root system of *Trifolium repens* L. is rod-shaped, with highly branching lateral processes. The bulk of the roots is located in the 40–50 cm soil layer for *Trifolium repens* L. creeping and its varieties are not demanding on soils, but they develop well on clay and loamy types, tolerate the proximity of groundwater better than other legumes (85–90 cm), and have high winter and frost resistance. *Trifolium repens* L. can accumulate both anions and cations effectively and can compensate for the soil enzyme activity loss caused by heavy metal [42].

Several authors have used *Trifolium repens* L. as test plants for studying the phytoextraction of heavy metals in the soil and climatic conditions of Russia, where there are large areas of soddy podzolic loamy soils [43,44]. High efficiency of clover in absorbing Ni, Cu, Zn was noted.

The seeds of the company "Turfline" LLC, Russian Federation, were used in the experiments.

### 3.1.2. Reactive and Preparators

- Universal soil "SELIGER-AGRO EXO" (of the company "Seliger Agro", Tver, Russia). The main characteristics of the soil are shown in Table 1.
- Nitrogen-phosphorus-potassium fertilizer brand "Antey" "Earth Force" according to RF Specification TU # 2186-002-38522882-2016. Fertilizer contains 21% nitrogen, 11% phosphorus ($P_2O_5$), 11% potassium.
- Universal preparation "Zavyaz" ("Orton" LLC, Pushkino, Russia), containing 5% mass sodium salts of (GA).
- Preparation "Kornevin" ("SELHOZEKOSERVICE" LLC, Moscow, Russia), containing IAA (in form 4 (indole-3yl) butyric acid) at a concentration of 5 g/kg.
- Pure $Ni(NO_3)_2 \cdot 6H_2O$ (nickel nitrate 6-aqueous) for analysis.
- Pure $CuSO_4 \cdot 5H_2O$ (copper (II) sulfate 5-water).
- Pure $Cd(NO_3)_2 \cdot 4H_2O$ (cadmium nitrate 4-aqueous).
- Na (FeEDDHA) in form of 0.1% solution prepared at the Laboratory of Institute for Chemical Reagents and High Purity Chemical Substances of National Research Center "Kurchatov Institute"—IREA.
- $Na_2EDTA$ prepared at the Laboratory of Institute for Chemical Reagents and High Purity Chemical Substances of the National Research Center "Kurchatov Institute"— IREA.
- $K_2HEDP$—an aqueous solution with a mass content of the target component of 28.3% was provided by the Laboratory of Institute for Chemical Reagents and High Purity Chemical Substances of National Research Center "Kurchatov Institute"—IREA.

**Table 1.** The main characteristics of the universal soil "SELIGER-AGRO EXO" [45].

| # | Characteristic Name | Characteristic Value |
|---|---|---|
| 1 | Packaging volume | 60 L |
| 2 | Origin of soil | Natural high-moor peat, neutralized with lime, with the addition of complex mineral fertilizer |
| 3 | The type of plants for which this type of soil is applicable | Ornamental, deciduous, herbaceous |
| 4 | Mass fraction of water | Up to 70% |
| 5 | Acidity (pH KCl) | 5–6 |
| 6 | Nitrogen content ($NH_4 + NO_3$) | 100–180 mg/L |
| 7 | Phosphorus content ($P_2O_5$) | 135–255 mg/L |
| 8 | Potassium content ($K_2O$) | 115–215 mg/L |
| 9 | Packaging weight | 18 kg |

*3.2. Methods*

The experiments were carried out following the ISO 22030: 2005 standard "Soil quality. Biological methods. Chronic toxicity in higher plants".

During the experiment, the vegetation pot was filled with universal soil with the addition of 237 mg of fertilizer containing 21% nitrogen, 11% phosphorus ($P_2O_5$), and 11% potassium. The soil was thoroughly mixed.

For modeling a simulated multi-metal-contaminated substrate, weighed portions of reagents containing heavy metals were added to the vegetation pots in an amount corresponding to five times the maximum permissible concentration (MPC) established in Russia in terms of content in the soil:

- $Ni(NO_3)_2 \cdot 6H_2O$—38.2 mg per 1 vegetation pot;
- $CuSO_4 \cdot 5H_2O$—23.18 mg per 1 vegetation pot;
- $Cd(NO_3)_2 \cdot 4H_2O$—15.65 mg per 1 vegetation pot.

All vegetation pots were planted with the *Trifolium repens* L. seeds at the rate of 20 seeds per vegetation pot.

Next, amendments were added to the individual vegetation pots according to Table 2. Split treatments were used for the chelating agents. The calculated dose of reagents was divided into five portions, which were injected sequentially and daily for five days.

**Table 2.** Description of variants of experiments in vegetation pots with heavy metals with and without amendments and with clean soil.

| Variant of Experiment | Experiment Description | Amendments | Application Time | Application Method |
|---|---|---|---|---|
| Clean soil | The experiment uses only universal soil and fertilizer | No amendments | - | - |
| Ni, Cu, Cd/Ni (no amendments) | The experiment uses only universal soil contaminated with heavy metals and fertilizer | No amendments | - | - |
| Ni, Cu, Cd/Ni + $Na_2EDTA$ | $Na_2EDTA$ is added to heavy metal contaminated universal soil with fertilizer | $Na_2EDTA$ (12.06 g of the reactive was diluted in 600 mL of distilled water) | From 20 to 25 days after planting the seeds | The solution was added in an amount of 20 mL by watering into each vegetation pot |

**Table 2.** *Cont.*

| Variant of Experiment | Experiment Description | Amendments | Application Time | Application Method |
|---|---|---|---|---|
| Ni, Cu, Cd/Ni + K$_2$HEDP | K$_2$HEDP is added to heavy metal contaminated universal soil with fertilizer | K$_2$HEDP (2 mL of 28.3% solution was diluted in 1 L of distilled water) | From 20 to 25 days after planting the seeds | The solution was added in an amount of 10–20 mL by watering into each vegetation pot |
| Ni, Cu, Cd/Ni + K$_2$HEDP + GA + IAA | K$_2$HEDP is added to heavy metal contaminated universal soil with fertilizer with exogenous treatments with PGRs (GA and IAA) | K$_2$HEDP (2 mL of 28.3% solution was diluted in 1 L of distilled water) | From 20 to 25 days after planting the seeds | The solution was added in an amount of 10–20 mL by watering into each vegetation pot |
| | | GA (0.2 g of universal preparation "Zavyaz" was diluted in 1 litre of distilled water) | 12, 20, and 28 days after planting the seeds | The solution was sprayed until the ground was moistened |
| | | IAA (0.7 g preparation "Kornevin" was diluted in 1 L of distilled water) | 12, 20, and 28 days after planting the seeds | The solution was added in an amount of 10 mL by watering into each vegetation pot |
| Ni, Cu, Cd/Ni + K2HEDP + GA + IAA + Na(FeEDDHA) | K$_2$HEDP is added to heavy metal contaminated universal soil with fertilizer with exogenous treatments with PGRs (GA and IAA) and Na (FeEDDHA) | K$_2$HEDP (2 mL of 28.3% solution was diluted in 1 L of distilled water) | From 20 to 25 days after planting the seeds | The solution was added in an amount of 10–20 mL by watering into each vegetation pot |
| | | GA (0.2 g of universal preparation "Zavyaz" was diluted in 1 L of distilled water) | 12, 20, and 28 days after planting the seeds | The solution was sprayed until the ground was moistened |
| | | IAA (0.7 g preparation "Kornevin" was diluted in 1 L of distilled water) | 12, 20, and 28 days after planting the seeds | The solution was added in an amount of 10 mL by watering into each vegetation pot |
| | | Na (FeEDDHA) solution of concentration 0.001% | 12, 20 and 28 days after planting the seeds | The solution was sprayed until the ground was moistened |

All variants of the experiments in Table 2 were conducted in triplicate.

The plants were dug up 31 days after planting the seeds. The moved plants were cleared of soil, washed with water, and divided into roots and shoots for further studies on the content of heavy metals.

Method for Determination of Metals in Plants

The determination method was based on the use of the autoclaved (with resistive heating) acid decomposition of the analyzed samples and subsequent analysis of the resulting solution by two multielement methods: Atomic emission spectrometry with inductively coupled plasma (AES–ICP) and mass spectrometry with inductively coupled plasma (MS–ICP). The AES–ICP method (iCAP-6500, Thermo Scientific, Waltham, MA, USA) allowed us to determine the contents of Li, B, Na, and Ba. The MS–ICP method (X-7, Thermo Electron, Waltham, MA, USA) allowed us to determine the contents of Li, Be, B, Sc, Ti, V, Cr, Mn, Co, Ni, Cu, Zn, Ga, As, Se, Rb, Sr, Y, Mo, Rh, Ag, Cd, Sn, Sb, Te, Cs, Ba, La, Ce, Pr, Nd, Sm, Eu, Gd, Tb, Dy, Ho, Er, Tm, Yb, Lu, Re, Ir, Pt, Au, Hg, Tl, Pb, Bi, Th, and U.

### 3.3. Statistical Analysis

Data are presented as the arithmetic mean and standard deviation or the coefficient of variation, and the data were statistically compared between groups using Fisher's test with Microsoft Office Excel 2007 software.

### 4. Results

On the 31st day of the experiment, after planting the *Trifolium repens* L. seeds in pots with a substrate contaminated with heavy metals, germination, and carrying out the planned treatments, the state of the seedlings was assessed. The seedlings in the vegetation pots contaminated with Ni, Cu, and Cd supplemented with $K_2HEDP$ (Figure 1b) looked almost identical to the seedlings in the vegetation pots contaminated with Ni, Cu, and Cd without the addition of amendments (Figure 1a). However, the vegetation pots contaminated with Ni, Cu, and Cd supplemented with $K_2HEDP$ + GA + IAA + Na(FeEDDHA) (Figure 1c) had visually larger biomasses than the seedlings in the vegetation pots contaminated with Ni, Cu, and Cd without the addition of amendments (Figure 1a). Compared to the rest of the samples, the plants in the pot with the addition of $Na_2EDTA$ (Figure 1d) looked wilted and pale with noticeable white patches, some leaves were yellowed, and there were signs of chlorosis. In the rest of the pots, the plants were green, and on some leaves, there were white blotches (see Table 3).

**Table 3.** The appearance of the *Trifolium repens* L. seedlings after experiments.

| Samples | Shoot Turgor | Shoot Growth | Signs of Chlorosis | Sheet Plates |
|---|---|---|---|---|
| Clean soil | High | Natural | No | Green |
| Ni, Cu, Cd (no amendments) | High | Minor oppression | No | Green |
| Ni, Cu, Cd + $Na_2EDTA$ | Low | Significant oppression | Yes | Pale green |
| Ni, Cu, Cd + $K_2HEDP$ | High | Significant oppression | No | Green |
| Ni, Cu, Cd + $K_2HEDP$ + GA + IAA | High | Average oppression | No | Green |
| Ni, Cu, Cd + $K_2HEDP$ + GA + IAA + Na(FeEDDHA) | High | Minor oppression | No | Green |

Green color: Highlight the difference.

The results of the study of the *Trifolium repens* L. seedlings after the experiments are shown in Table 4 and Figure 2.

**Table 4.** Weight and metal content in shoots and roots of the *Trifolium repens* L. seedlings after experiments.

| Variant of Experiment | Shoots | | | | Roots | | | |
|---|---|---|---|---|---|---|---|---|
| | Mass [1] [g] | Concentration [2] [μg/g] | | | Mass [g] | Concentration [μg/g] | | |
| | | Ni | Cu | Cd | | Ni | Cu | Cd |
| Clean soil | 0.278 | 6.4 | 2.1 | 0.19 | 0.024 | 25.3 | 5.6 | 1.4 |
| Ni, Cu, Cd (no amendments) | 0.24 | 9.1 | 6.7 | 10.1 | 0.014 | 19 | 17.4 | 189 |
| Ni, Cu, Cd + $Na_2EDTA$ | 0.10 | 54.3 | 40.4 | 32.9 | 0.009 | 27 | 41 | 56.7 |
| Ni, Cu, Cd + $K_2HEDP$ | 0.14 | 13.5 | 7.6 | 3.7 | 0.007 | 49.9 | 27.5 | 123 |
| Ni, Cu, Cd + $K_2HEDP$ + GA + IAA | 0.18 | 7.8 | 5.2 | 10.4 | 0.012 | 28 | 13.3 | 70.9 |
| Ni, Cu, Cd + $K_2HEDP$ + GA + IAA + Na(FeEDDHA) | 0.23 | 9.7 | 4.2 | 8.5 | 0.010 | 26 | 12.6 | 144 |

[1] The relative standard deviation for mass did not exceed 0.05 [g]. [2] The relative standard deviation for all concentration elements (Ni, Cu, Cd) did not exceed 0.3 [μg/g].

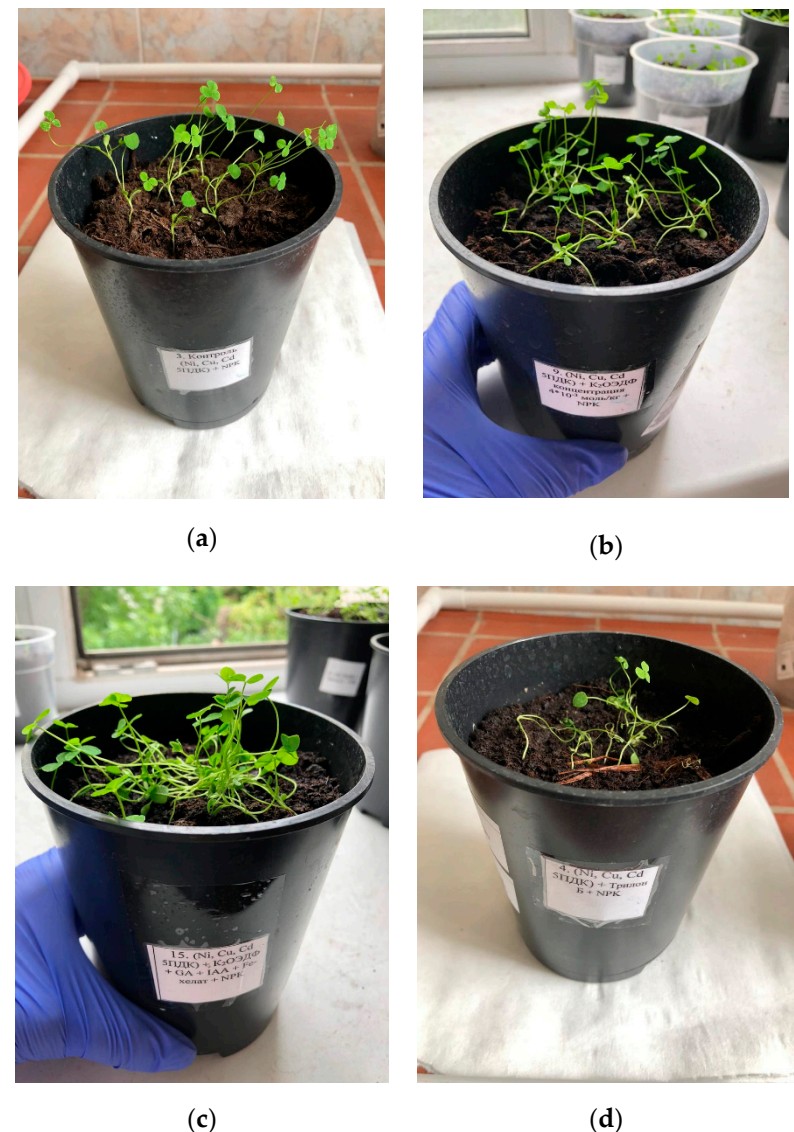

**Figure 1.** The *Trifolium repens* L. seedlings on the 31st day of the experiment after planting seeds: (**a**) in vegetation pots contaminated with Ni, Cu and Cd without the addition of amendments; (**b**) in vegetation pots contaminated with Ni, Cu and Cd supplemented with $K_2HEDP$; (**c**) in vegetation pots contaminated with Ni, Cu иCd supplemented with $K_2HEDP$ + GA + IAA + Na(FeEDDHA); (**d**) in vegetation pots contaminated with Ni, Cu иCd supplemented with $Na_2EDTA$.

The translocation coefficient (TF) of Ni roots to shoots, calculated as the ratio of its concentrations in the shoots to roots, was the highest in the *Trifolium repens* L. seedlings when using $Na_2EDTA$ (Figure 3).

As a result, the maximum amount of Ni accumulated in the *Trifolium repens* L. seedlings from three vegetation pots with the addition of $Na_2EDTA$ was 5.4 µg, and with the addition of $K_2HEDP$ with growth regulators and iron chelate, this was 2.5 µg; in the other cases, the Ni content was slightly lower. At the same time, in the samples of *Trifolium repens* L. seedlings, an increase in Ni uptake in the shoots was observed in the presence of $Na_2EDTA$, as well as in the roots in the presence of $K_2HEDP$. The use of PGRs and $Na_2EDTA$ had a pronounced negative effect on the ability of the *Trifolium repens* L. seedlings to accumulate Ni in their roots, but had a positive effect on TF.

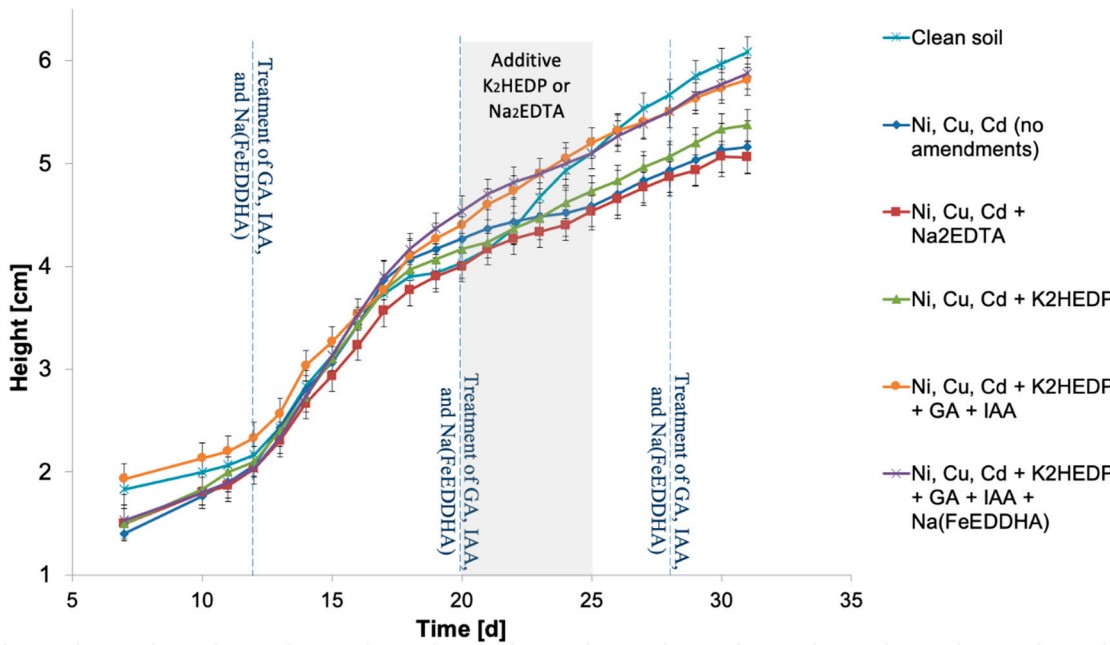

**Figure 2.** Dependence of the *Trifolium repens* L. seedlings' growth on the introduction of various amendments.

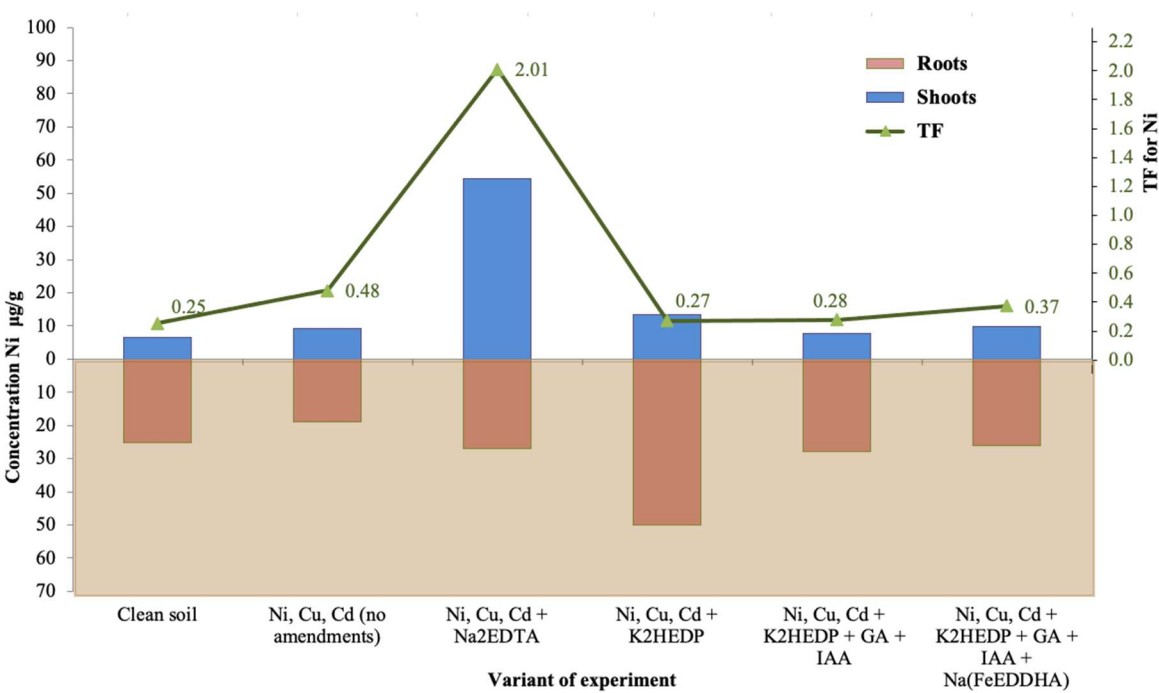

**Figure 3.** Analysis of Ni transport in shoots and roots of the *Trifolium repens* L. seedlings using various amendments.

A slightly different situation was observed for the transfer of Cu to the organs of the *Trifolium repens* L. seedlings (Figure 4).

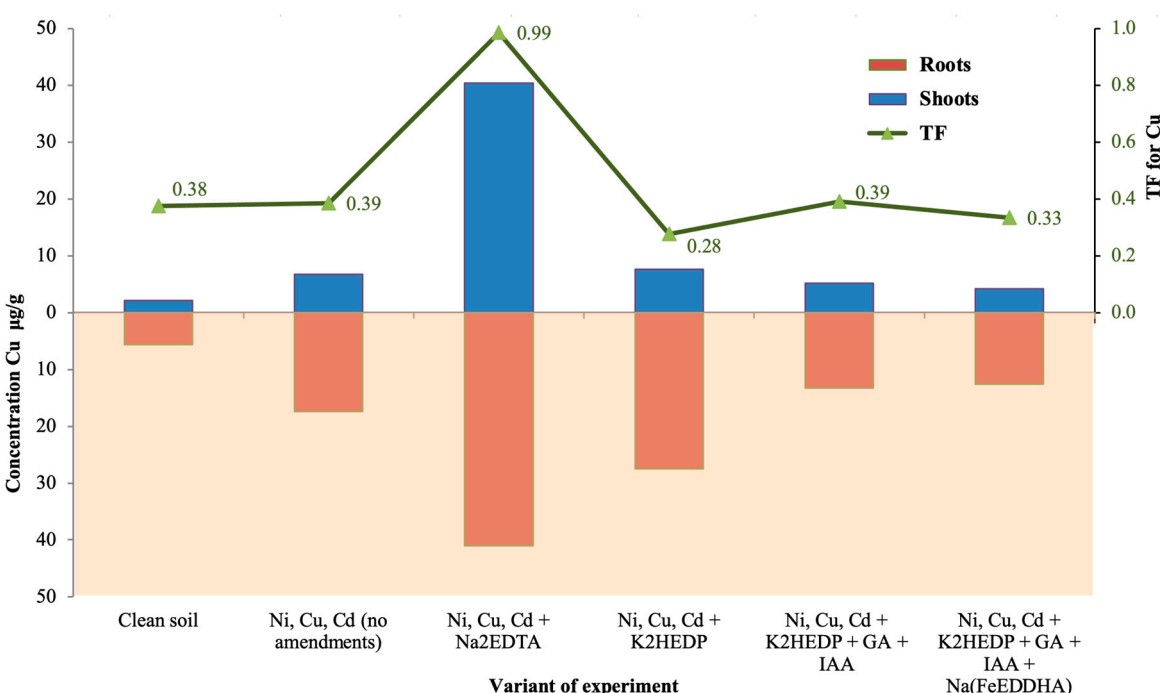

**Figure 4.** Analysis of Cu transport in shoots and roots of the *Trifolium repens* L. seedlings when using the various amendments.

Figure 4 shows an almost equal distribution of Cu in the shoots and roots of the *Trifolium repens* L. seedlings in the experiments with $Na_2EDTA$, which, similarly to Ni, was significantly higher than in the other samples. The actions of the PGRs and $Na_2EDTA$ were similar to that of Ni. The use of $K_2HEDP$ without PGRs and $Na_2EDTA$ caused an increase in the Cu content in the roots of the *Trifolium repens* L. seedlings, which is not desirable since it negatively affected the value of the TF.

The maximum amount of Cu accumulated in total in the *Trifolium repens* L. seedlings from three vegetation pots with the addition of $Na_2EDTA$ was 4.2 µg, followed by plants grown in contaminated soil without amendments, which, in total, were able to accumulate 1.9 µg of Cu.

Cd was able to accumulate in the roots of the *Trifolium repens* L. seedlings at higher concentrations than Ni and Cu (Figure 5).

The accumulation of Cd in the roots of *Trifolium repens* L. was especially intense in the absence of any amendments. The use of $Na_2EDTA$ reduced the concentration of metal in the roots of the *Trifolium repens* L. seedlings by three times and simultaneously increased its content in the shoots by three times, which, in turn, led to a 10-fold increase in TF. The addition of $K_2HEDP$ led to a 2-fold decrease in the content of cadmium in both the roots and shoots of the *Trifolium repens* L. seedlings., in comparison to its content in the plant organs growing under conditions without the use of chemical amendments, as well as led to a decrease in TF. The additional introduction of PGRs and $Na_2EDTA$ allowed a slight increase in both the metal content in the plant organs and TF.

The maximum amount of Cd (5.1 µg in three pots in total) was found in the *Trifolium repens* L. seedlings grown in contaminated soil without the use of any amendments. Furthermore, approximately the same amount was found in plants grown in pots with the addition of $Na_2EDTA$ (3.6 µg) and $K_2HEDP$ with the addition of PGRs and $Na_2EDTA$ (3.4 µg). In the other cases, the amount of absorbed Cd was significantly lower.

A summary of the results on the accumulation of Cd, Cu, and Ni in the shoots and roots of the *Trifolium repens* L. seedlings, taking into account both the mass of plant organs and the concentration of the metals in them, is shown in Figure 6.

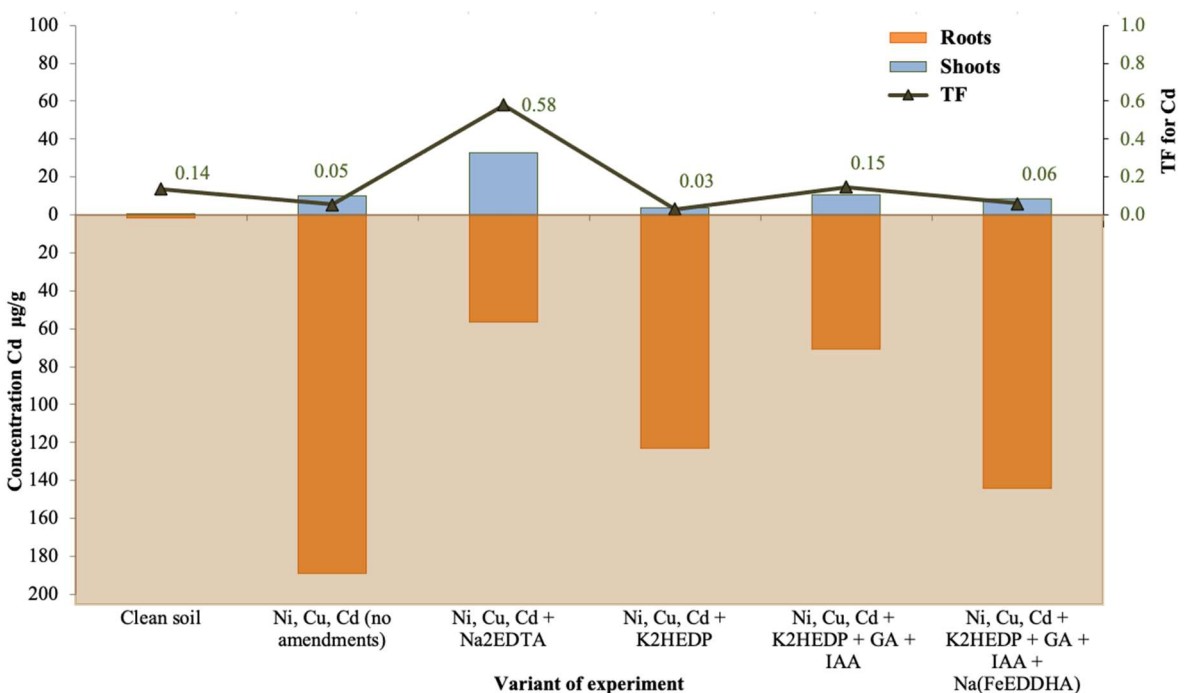

**Figure 5.** Analysis of Cd transport in shoots and roots of the *Trifolium repens* L. seedlings when using the various amendments.

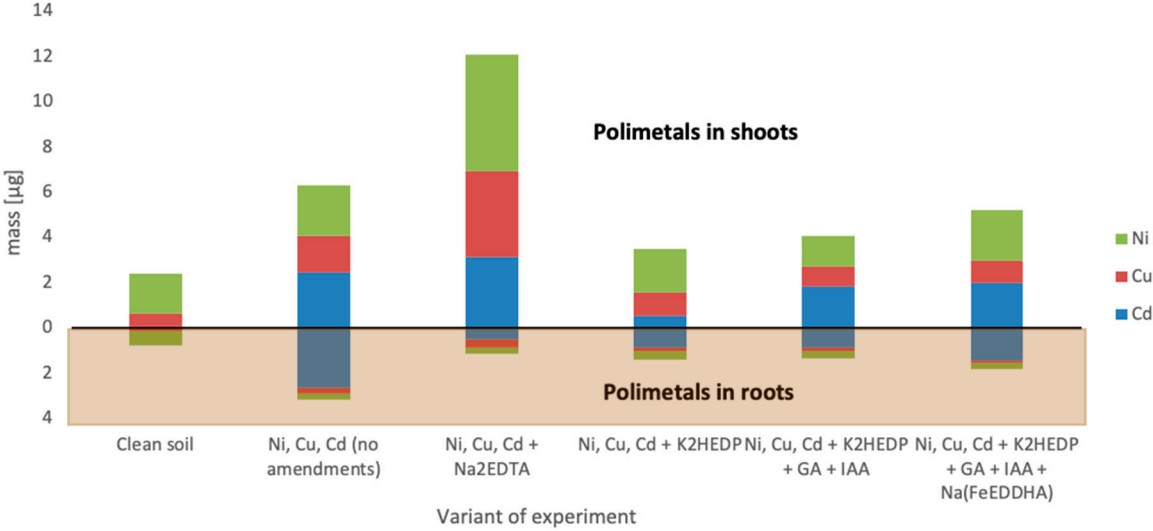

**Figure 6.** Summary data on the accumulation of Cd, Cu, and Ni of the *Trifolium repens* L. seedlings shoots and roots depending on the use of various amendments.

Comparative results of the behavior of Ni in the composition of the polymetallic pollution from pure Ni pollution are shown in Figure 7.

Figure 8 shows that the accumulation of Ni in the organs of the *Trifolium repens* L. seedlings was higher in the absence of other metallic contaminants.

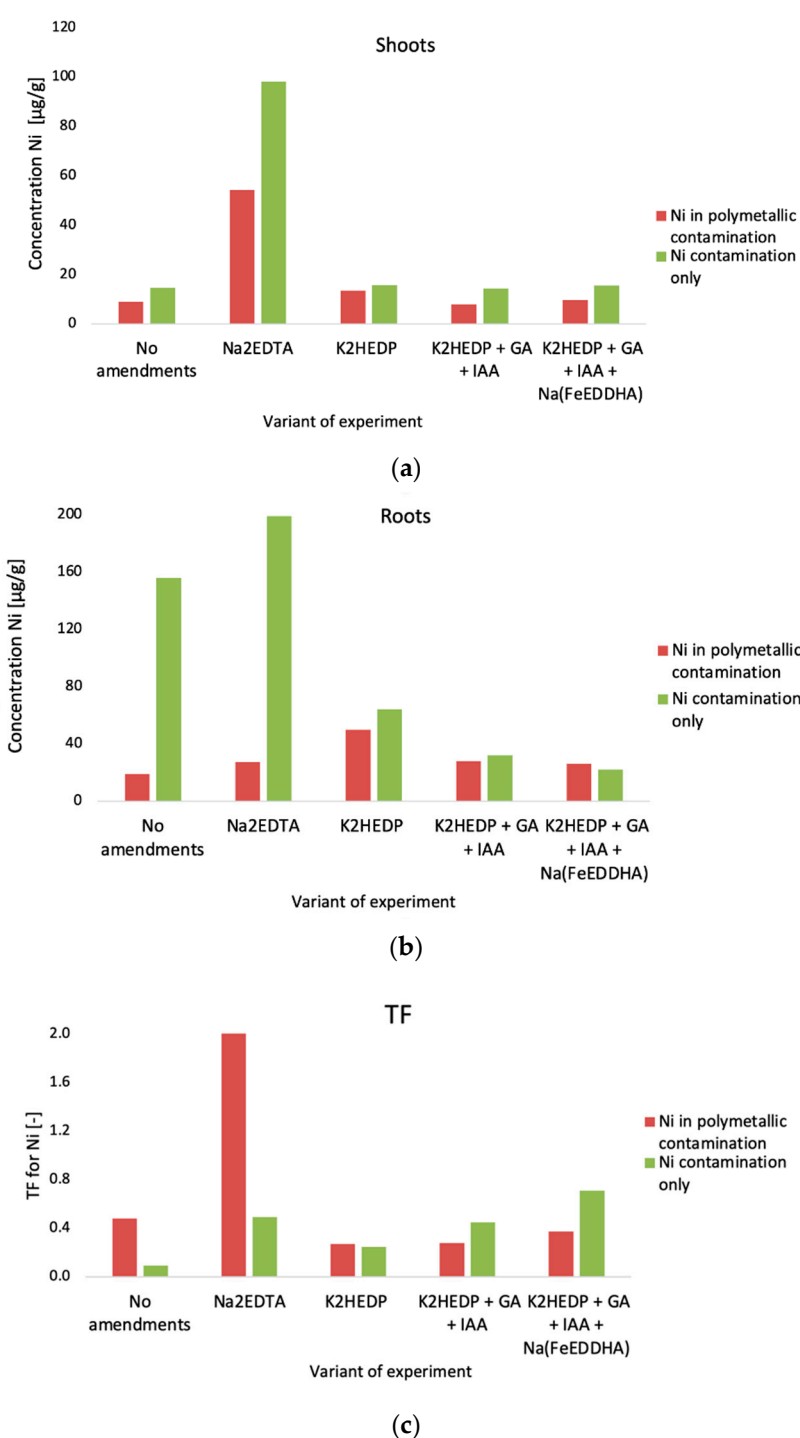

**Figure 7.** Summary data on Ni accumulation in the *Trifolium repens* L. seedlings in polymetallic contamination and pure nickel contamination depending on the use of various amendments: (**a**) in shoots; (**b**) in roots; (**c**) translocation coefficient (TF).

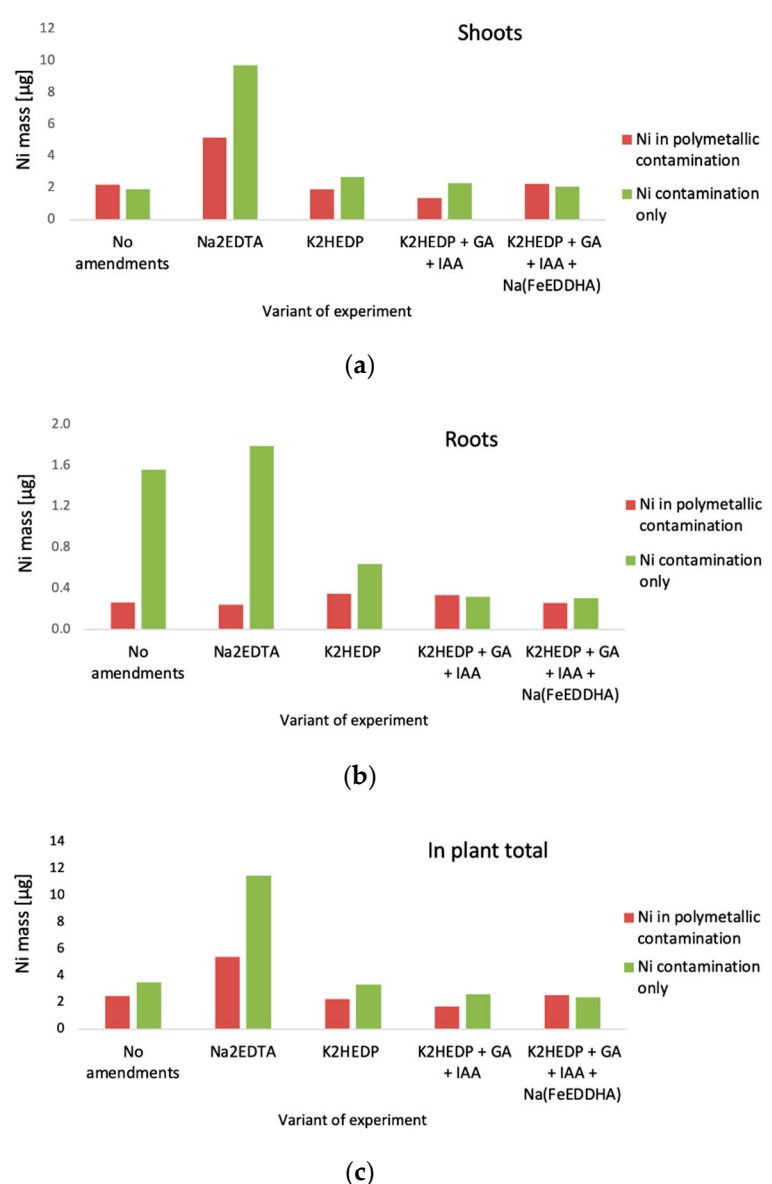

**Figure 8.** Summary data on Ni accumulation in the *Trifolium repens* L. seedlings, depending on the use of various chemical amendments and the type of pollution: (**a**) in shoots; (**b**) in roots; (**c**) in plant total.

Figure 9 shows the results of the elemental analysis of the *Trifolium repens* L. seedlings grown in contaminated soils with various amendments with PGRs and Na (FeEDDHA). The variants of the experiments in which there was an excess of one or another element in the *Trifolium repens* L. seedlings in comparison to plants grown in clean soils are marked in green. The cases with an excess of two or more times are marked in bright green. The experiments in which the content of the assessed element in the *Trifolium repens* L. seedlings was lower than in samples grown in clean soil are marked in brown.

| Shoots | | | | | | Roots | | | | | |
|---|---|---|---|---|---|---|---|---|---|---|---|
| Ni, Cu, Cd (no amendments) | Ni, Cu, Cd + Na$_2$EDTA | Ni, Cu, Cd + K$_2$HEDP | Ni, Cu, Cd + K$_2$HEDP + GA + IAA | Ni, Cu, Cd + K$_2$HEDP + GA + IAA + Na(FeEDDHA) | | Ni, Cu, Cd (no amendments) | Ni, Cu, Cd + Na$_2$EDTA | Ni, Cu, Cd + K$_2$HEDP | Ni, Cu, Cd + K$_2$HEDP + GA + IAA | Ni, Cu, Cd + K$_2$HEDP + GA + IAA + Na(FeEDDHA) | |
| 1 | 1.2 | 1.1 | 1 | 1 | Mg | 1.2 | 1.6 | 1.5 | 1.8 | 2.1 | Biogenic macronutrients |
| 0.9 | 1 | 0.9 | 0.9 | 1 | P | 1.1 | 1.1 | 1 | 1.2 | 1.4 | |
| 1.1 | 1.2 | 1.1 | 1 | 1.1 | S | 1.3 | 1.4 | 1.5 | 1.3 | 1.6 | |
| 1.1 | 0.8 | 0.9 | 1 | 1.1 | K | 1.4 | 1.5 | 1.1 | 1.1 | 1.4 | |
| 1 | 1 | 1.2 | 1 | 1 | Ca | 1.6 | 2.1 | 1.4 | 1.5 | 2.1 | |
| 1.7 | 13.9 | 1.5 | 1.3 | 1.4 | Na | 2.4 | 20.4 | 3.5 | 1.4 | 2.7 | |
| 1.3 | 4.5 | 1.1 | 1.1 | 1.1 | Fe | 2.3 | 2.4 | 1.3 | 1.2 | 1.5 | |
| 0.6 | 1.5 | 3.6 | 0.8 | 1.1 | Cr | 2.4 | 2.5 | 1.5 | 1.6 | 2.1 | Biogenic trace elements |
| 1 | 1.2 | 0.9 | 0.9 | 0.9 | Mn | 3.8 | 1.1 | 1.7 | 1.8 | 5.3 | |
| 0.5 | 0.9 | 0.7 | 0.7 | 0.6 | Co | 1.4 | 1.1 | 1.1 | 1.3 | 1.4 | |
| 1.3 | 0.9 | 1 | 1.2 | 1.1 | Zn | 1.7 | 0.9 | 1.3 | 1.9 | 1.7 | |
| 2.4 | 1.9 | 1.6 | 3.5 | 2.7 | Mo | 0.7 | 0.8 | 1 | 1.2 | 1.1 | |
| 1.1 | 1 | 0.8 | 0.8 | 0.8 | B | 1 | 1.5 | 0.9 | 1 | 0.9 | |
| 1.4 | 4.5 | 1.5 | 2.2 | 3.3 | Al | 1.9 | 2.3 | 1.3 | 1.3 | 1.5 | |
| 2.3 | 2.3 | 2.1 | 1.5 | 1.6 | Ti | 2.2 | 3.1 | 2 | 2 | 2.1 | |
| 1 | 2 | 1 | 1 | 1.3 | Ga | 1.8 | 1.8 | 1 | 0.9 | 1.5 | |
| 1.2 | 3.1 | 1.6 | 1.5 | 1.2 | As | 0.8 | 0.6 | 1.1 | 1.2 | 1 | |
| 1 | 0.7 | 0.8 | 1 | 1 | Rb | 1.7 | 1.6 | 1.2 | 1.1 | 1.5 | |
| 0.9 | 1.1 | 1.1 | 1 | 1 | Sr | 1.4 | 1.8 | 1.3 | 1.3 | 1.6 | |
| 1 | 9.3 | 1 | 1 | 1.3 | Y | 2.1 | 3.9 | 1.4 | 1.5 | 1.7 | |
| 2.3 | 3.7 | 6.7 | 1.4 | 2.3 | Ag | 1.7 | 2.5 | 4.7 | 1 | 2.7 | |
| 1 | 1 | 1 | 1 | 1 | Sn | 1.9 | 1 | 2.2 | 1 | 2.5 | |
| 1 | 1 | 1 | 1 | 1 | Sb | 1 | 1 | 1.4 | 1 | 2.5 | |
| 1 | 0.9 | 1 | 0.7 | 0.9 | Cs | 1.8 | 1.4 | 1.2 | 0.9 | 1.1 | |
| 1 | 1.5 | 1.2 | 1.2 | 1.1 | Ba | 1.8 | 2.3 | 1.7 | 1.7 | 2.8 | |
| 1.3 | 4.1 | 1.3 | 2.5 | 4.7 | La | 1.4 | 2.8 | 1 | 0.9 | 1.2 | |
| 0.9 | 3.8 | 1.1 | 1.6 | 2.6 | Ce | 1.3 | 2.8 | 1 | 0.9 | 1.1 | |
| 1.5 | 5.8 | 1.4 | 3.4 | 6 | Pr | 1.8 | 3.3 | 1.3 | 1.4 | 1.4 | |
| 1.4 | 5.9 | 1.3 | 2.7 | 5.1 | Nd | 2 | 3.1 | 1.3 | 1.3 | 1.5 | |
| 1.2 | 4.7 | 1.4 | 2.7 | 3.7 | Sm | 1.8 | 2.5 | 1.2 | 1.3 | 1.7 | |
| 1 | 1 | 1 | 1 | 1.2 | Eu | 2.1 | 3.2 | 1.1 | 0.7 | 0.7 | |
| 0.6 | 3.8 | 1.2 | 1.3 | 1.9 | Gd | 1.7 | 2.5 | 1.1 | 1.1 | 1.3 | |
| 1 | 3 | 1 | 1.6 | 1 | Tb | 2.3 | 3.3 | 1.3 | 1.8 | 1.4 | |
| 1 | 5.6 | 1 | 1 | 1.3 | Dy | 2.1 | 3.5 | 1.4 | 1.4 | 1.4 | |
| 1 | 5.9 | 1 | 1 | 1.2 | Ho | 1.6 | 3.9 | 1.2 | 2.1 | 2 | |
| 1 | 4.6 | 1 | 1 | 1 | Er | 1.2 | 2 | 0.9 | 1 | 1.3 | |
| 1 | 2.4 | 1 | 1 | 1 | Tm | 2.1 | 1 | 1 | 1 | 1 | |
| 1 | 3.5 | 1 | 1 | 1 | Yb | 1.1 | 2.4 | 1 | 1.6 | 1.6 | |
| 2.1 | 1.7 | 1.3 | 1.5 | 1.3 | Hg | 2 | 2.3 | 1.9 | 0.9 | 2.9 | |
| 1.5 | 1.4 | 1 | 1.4 | 1.4 | Tl | 1.4 | 1.1 | 1.1 | 0.9 | 1.3 | |
| 1.9 | 5.2 | 2.2 | 2.4 | 2 | Pb | 2.3 | 3.6 | 2.2 | 1.4 | 3.4 | |
| 1 | 1 | 2.2 | 1 | 1 | Bi | 1 | 1 | 1.3 | 1 | 1 | |
| 1 | 1.7 | 1 | 1.6 | 2.9 | Th | 1.8 | 1.7 | 1 | 1.2 | 1 | |
| 2.1 | 1.9 | 1 | 1.3 | 1.5 | U | 2 | 1.8 | 1.2 | 1.3 | 1.4 | |
| 1 | 0.7 | 1.1 | 0.8 | 0.8 | Li | 1.9 | 2 | 1.5 | 1 | 1.7 | |

**Figure 9.** Evaluation of changes in the content of elements in the *Trifolium repens* L. seedlings with plants grown in clean soils.

## 5. Discussion

The data presented in Table 4 clearly show the negative effect of increased concentrations of heavy metal ions on the development of the *Trifolium repens* L. seedlings. Thus, contamination of the substrate with Ni, Cu, and Cd ions at the level of 5 MPC led to a decrease in biomass by 14%. The application of amendments (Na$_2$EDTA and K$_2$HEDP) to increase the degree of absorption of the heavy metals by plants had an even stronger effect, especially in the case of Na$_2$EDTA (more than two times). The appearance of plants (Figure 1 and Table 3) also indicated a pronounced negative effect of Na$_2$EDTA.

Treatment with the PGRs (GA and IAA phytohormone complex) had a significant compensatory effect on the biomass of shoots and roots (see Table 4). Moreover, the addition of iron chelates to the processing technology made it possible to further increase the mass of shoots and almost completely compensated for the suppression of shoots caused by the use of a chelating reagent—namely, K$_2$HEDP (0.23 vs. 0.14 g). Thus, additional treatment

with exogenous phytohormones, together with iron chelate, showed the best practical results in terms of increasing the biomass of phytoextractor plants.

The maximum increase in the absorption of heavy metals (Ni, Cu, and Cd) was shown by the use of $Na_2EDTA$. A significant increase in absorption was observed for all three metals in the shoots. The concentrations of Ni, Cu, and Cd increased by 5.96, 6.03, and 3.26 times, respectively (see Figures 3–5) compared to the Ni, Cu, and Cd concentrations in the experiment without amendments. Additionally, in the roots, significant increases in the concentrations of Ni and Cu were also recorded (1.42 and 2.36 times, respectively) in the same experiment. However, the Cd content (Figure 5) in the sample with $Na_2EDTA$ was significantly lower than that in the root part of the sample without amendment—i.e., 3.33 times. The reason for this fact may lie in the specific relation to Cd of the *Trifolium repens* L. The data obtained allow us to make an unambiguous conclusion about the positive effect of $Na_2EDTA$ on the studied polymetallic contamination: this amendment unambiguously promotes the accumulation of polymetals in shoots of *Trifolium repens* L. seedlings.

Compared to the use of $Na_2EDTA$, $K_2HEDP$ showed a completely different effect on the absorption of metal ions. On the one hand, there were no significant increases in the concentrations of Ni, Cu, and Cd in the shoots in the experiment with $K_2HEDP$ as compared to the variant without amendments. On the other hand, significant increases in the Ni and Cu concentrations in the roots were recorded—by 2.63 and 1.58 times compared to the experiment without amendments. In the plant phytoextractor *Trifolium repens* L., there was an increase in the "exclusion" strategy in the experiment with $K_2HEDP$, when heavy metals were concentrated in the roots and the transfer to other organs was limited, thus resulting in resistance to high levels of increased pollution. This fact, to some extent, could be explained by the specific properties of the ligand—oxyethylidene diphosphonic acid (HEDP)—belonging to the class of bisphosphonates. Based on its structure, HEDP is a complete analogue of natural pyrophosphates and is capable of participating in cellular reactions, regulating ionic calcium and phosphorus metabolism. There is evidence that HEDP can influence the stabilization of cell membranes by interacting with the ligands of membrane proteins and by integrating into membrane structures. However, the metabolism of HEDP in plants has not been studied in detail; therefore, an exhaustive explanation of the effect found during this experiment is fraught with certain difficulties. Based on the data obtained, a reliable conclusion can be drawn about the specific effect of the HEDP ligand on the absorption of metal ions by *Trifolium repens* L.

The use of the combinations of $K_2HEDP$ + GA + IAA and $K_2HEDP$ + GA + IAA + Na (FeEDDHA) did not have any significant additional effect on the increase in absorption, but also slightly enhanced the opposite effect, i.e., inhibition, which is an uncharacteristic property for the commonly used carboxyl-containing inducer class of complexons.

To assess the reliability of the findings, the cumulative accumulation of metals (Figure 6) in the roots and shoots was evaluated. On the one hand, such an assessment fully confirmed the earlier conclusions regarding $Na_2EDTA$ and $K_2HEDP$. The total accumulation of the whole seedlings using $Na_2EDTA$ reached the value of 13.88 µg, which is 48% more than without the application of the amendment. Samples treated with $K_2HEDP$ showed a general effect that, on the contrary, can be characterized as inhibitory or stabilizing. However, the complex treatments $K_2HEDP$ + GA + IAA and $K_2HEDP$ + GA + IAA + Na (FeEDDHA), due to the increase in biomass yield and the compensatory mechanism, made it possible to obtain total bioaccumulation values of 11% and 44% more than that of the samples with $K_2HEDP$ alone.

To compare the behavior of Ni in the composition of polymetallic soil contamination and contamination of only Ni, similar experiments were carried out, but only with the participation of Ni as a monopollutant (Figures 7 and 8). The absolute Ni content in organs of the *Trifolium repens* L. seedlings, according to three parallel experiments, was higher if the soil was not additionally contaminated with other heavy metals. Moreover, both in the case of polymetallic contamination and in the case of purely nickel contamination, the

positive role of $Na_2EDTA$ is clear, and the effect of using $K_2HEDP$ both with PGRs and Na (FeEDDHA), and without them, is insignificant.

Figure 9 shows that the effect of $K_2HEDP$ on plants was significantly lower than the changes that occurred in the elemental composition of plants when using $Na_2EDTA$ or when grown in soils contaminated with polymetals without amendments. In the latter case, in particular, the translocation from the roots to leaves of such biogenic trace elements as Co and Cr decreased.

## 6. Conclusions

As a result of the experimental studies, data were obtained on the absorption efficiency of polymetals (Ni, Cd, and Cu,) by *Trifolium repens* L. seedlings, taking into account the introduction of various amendments: $Na_2EDTA$ and $K_2HEDP$ (with and without PGRs and Na(FeEDDHA)). Moreover, the results obtained were compared to the results of similar experiments on the phytoextraction of Ni from soils containing only Ni contamination.

In the course of this research, it was found that, concerning heavy metals, treatment of the soil substrate with a phosphorus-containing chelating agent (i.e., $K_2HEDP$) led to a transformation of the absorption mechanism in plants and a restriction of the intake of metals. From an application perspective, the weakened transport of heavy metals into the shoots is preferable for plants, including cultivated ones, when an increased content of the latter in terrestrial organs is undesirable.

A very definite positive contribution to the increase in the biomass and state of plants was made by the additional treatment of the shoots of *Trifolium repens* L. seedlings with PGRs and iron chelates. Further expansion deserves a study of the combined use of a complex of exogenous phytohormones and iron chelates in combination with carboxyl-containing chelators.

The results of this study confirmed that $Na_2EDTA$ has a pronounced negative effect on the growth and development (organ mass) of plants (this can be seen in the example of *Trifolium repens* L. seedlings). Additionally, $Na_2EDTA$ has a pronounced effect on the elemental composition of plants, which has a nonpositive effect on their viability. These facts, as well as the relative stability of $Na_2EDTA$ in the environment and its low complexation capacity at pH > 8, makes it unsuitable for phytoextraction, for example, at MSW landfill sites.

Taking into account the obtained results, the authors plan to conduct testing for other plant species that can be used for the phytoextraction of heavy metals on the territory of the Russian Federation, as well as for other metals in the composition of polymetallic contaminants. The authors also plan to conduct a comparative assessment of the effectiveness of $K_2HEDP$ and Na (FeEDDHA) of soils with pHs > 9.

Additionally, it is planned to conduct a long-term study in the field, since the level of the phytoextraction of metals by plants may have certain differences throughout their life cycles, and the fact that $K_2HEDP$ with PGRs and Na (FeEDDHA) has a positive effect on their growth and development may eventually lead to more pronounced advantages when using this composition in comparison to $Na_2EDTA$.

**Author Contributions:** Conceptualization, A.M. and E.N.; methodology, E.N.; formal analysis, T.A. and K.P.; investigation, T.A. and K.P.; writing—original draft preparation, A.M.; writing—review and editing, A.M. and E.N. All authors have read and agreed to the published version of the manuscript.

**Funding:** This research was funded by the Mendeleev University of Chemical Technology of Russia ac-cording to research project # 3-2020-039.

**Institutional Review Board Statement:** Not applicable.

**Informed Consent Statement:** Not applicable.

**Conflicts of Interest:** The authors declare no conflict of interest.

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
