# Peer review of "The Improved Phytoextraction of Heavy Metals and the Growth of Trifolium repens L.: The Role of K2HEDP and Plant Growth Regulators Alone and in Combination"

_sustainability, doi:10.3390/su13052432_

Round 1
Reviewer 1 Report
The manuscript of Makarova et al. deals with the effects of chelating agent and plant growth regulators in the context of phytoextraction of metal contaminated soil by Trifolium Repens L.
Definitely, there is an important contribution to the research field through this manuscript.
Comments and suggestions to the manuscript:
- The manuscript title should be reconstructed towards greater content potential, accuracy, and clarity.
- It is necessary to provide a definition of the term "amendments" and in more detail about their functional spectrum and application.
- The abstract needs to be substantially optimized, making it more conceptual and scientifically sounding.
- The whole of the manuscript text must be rewritten in terms of greater conceptualization, consistency, accuracy, and clarity.
- English needs to be greatly improved.
- It is necessary to provide a more reasoned rationale why the authors chose Sustainability as the target journal and on what grounds the concept of the manuscript corresponds to the goals and scopes of this journal.
- Why was Trifolium repens L. chosen as the preferable accumulator plant?
- How conceptually new is the authors' hypothesis formulated in the introduction? Substantiation is needed.
- It is also necessary to provide more informative data on the statistical processing of the obtained experimental data.
- A deeper analysis of the results is definitely needed in the context of a discussion of the other works on close and related topics. The latter should be presented in a wider spectrum.
Author Response
The authors are grateful to the reviewer for useful comments. We have highlighted in red the main corrections in the text of the manuscript.
1. The manuscript title should be reconstructed towards greater content potential, accuracy, and clarity.
The title of the article has been changed to
'Investigate Efficiency K2HEDP with Plant Growth Regulators for Phytoextraction of Heavy Metals by Trifolium Repens L.'
2. It is necessary to provide a definition of the term "amendments" and in more detail about their functional spectrum and application.
The following paragraph has been added to the text.
One of the effective ways to successfully solve the problems of increasing the solubility and bioavailability of metal ions in the soil substrate and increasing the potential of plants to accumulate pollutants in terrestrial organs is assisted phytoextraction using various functional amendments that enhance the absorption of metal ions by plants and increase the efficiency of the extraction process as a whole [14]. The best-studied amendments are chelating reagents. Chelating reagents are interacting with heavy metal cations and forming water-soluble complex compounds. This increase metals bioavailability.
3. The abstract needs to be substantially optimized, making it more conceptual and scientifically sounding.
Abstract has been rewritten.
'Heavy metals are among the most widespread pollutants of soil. Phytoextraction technology is used to solve the problem of multi-metal-contaminated soil. The efficiency of this process can be increased by introducing various amendments. A soil amendment is any material added to a soil to improve its physical properties, such as water retention, permeability, water infiltration, drainage, aeration and structure. Some chemical amendments for enhanced phytoextraction as amino polycarboxylates chelators can be hazardous to the environment and perform poorly at pH> 8. The potassium salt of hydroxyethylidene diphosphonic acid (K2HEDP) is suggested as a chemical amendment for enhanced phytoextraction. It works in a wider pH range and does not have a toxic effect on plants. The effect of K2HEDP on phytoextraction by Trifolium repens L. seedlings of Cd, Ni, and Cu was studied in this work. The results of the study confirmed that amino polycarboxylates chelators on example sodium salt of ethylene diamine tetraacetic acid (Na2EDTA) has a pronounced negative effect on the growth and development (organ mass) of Trifolium repens L. seedlings. K2HEDP, proposed by the authors instead of Na2EDTA, showed significantly lower efficiency in trials on the Trifolium repens L. seedlings. However, it should be noted its pronounced positive effect on plant growth and development, which was further enhanced by the use of plant growth regulators (PGRs) and with iron chelate. The highest was the efficiency of K2HEDP with PGRs and iron chelate for phytoextraction of Cd.'
4. The whole of the manuscript text must be rewritten in terms of greater conceptualization, consistency, accuracy, and clarity.
The comment was taken into account and explanations were introduced in the text.
5. English needs to be greatly improved.
The text was submitted for verification by a native British English speaker
6. It is necessary to provide a more reasoned rationale why the authors chose Sustainability as the target journal and on what grounds the concept of the manuscript corresponds to the goals and scopes of this journal.
This journal publishes an article on phytoremediation, for example:
- Ahila, K.G.; Ravindran, B.; Muthunarayanan, V.; Nguyen, D.D.; Nguyen, X.C.; Chang, S.W.; Nguyen, V.K.; Thamaraiselvi, C. Phytoremediation Potential of Freshwater Macrophytes for Treating Dye-Containing Wastewater. Sustainability 2021, 13, 329.
- Leudo, AM; Cruz, Y .; Монтойя-Руис, С .; Дельгадо, MdP; Салдарриага, Дж. Ф. Фиторемедиация ртути с помощью Lolium perenne -Mycorrhizae в загрязненных почвах. Устойчивое развитие 2020 , 12 , 3795.
- Etc.
7. Why was Trifolium repens L. chosen as the preferable accumulator plant?
The following paragraph has been added to the article:
The Trifolium repens L. (the white clover) seedlings. Clover is a very common wild-growing crop in the territory of the Russian Federation. In total, there are more than 70 growing species of clover within the country. Trifolium repens L., belonging to leguminous herbaceous crops, is kept in the herbage for 2-3 years. The root system of Trifolium repens L. is rod-shaped, with highly branching lateral processes. The bulk of the roots is located in the soil layer 40-50 cm Trifolium repens L creeping and its varieties are not demanding on soils, but they develop well on clay and loamy types, tolerate the proximity of groundwater better than other legumes (85-90 cm), has a high winter and frost resistance. Trifolium repens L. can accumulate both anions and cations effectively. Trifolium repens L. can compensate for soil enzyme activity loss caused by heavy metal [48].
Several authors used Trifolium repens L. as test plants in the study of phytoextraction of heavy metals in the soil and climatic conditions of Russia, where large areas are soddy podzolic loamy soils [49, 50]. High efficiency of clover in absorbing Ni, Cu, Zn was noted.
8. How conceptually new is the authors' hypothesis formulated in the introduction? Substantiation is needed.
Added to the introduction:
'But the choice and use of amendments are justified for each specific case since not all plants react in the same way and also depends a lot on the combination of external soil and climatic factors. Further development and improvement of zero-impact phytoextraction technology are closely related to the search for new mobilizing agents (chelators) and effective combinations of various amendments, which in combination with each other can demonstrate the best result.
Therefore, within the framework of this study of phytoextraction of soils with polymetallic contamination, it was of interest to solve the following problems:
- Testing as a new potential chemical inducer of a compound from the class of bisphosphonates - synthetic phosphorus-containing complexones. The organophosphorus, such as HEDP (hydroxyethylidene diphosphonic acid) are also capable of forming stable water-soluble complex compounds with many heavy metals, but in a much wider pH range. As an analogy of natural pyrophosphates, HEDP is involved in more than 60 biochemical cellular reactions by regulating ionic calcium and phosphorus exchange. And organophosphorus is considerably less toxic to living systems and organisms than carboxyl-containing complexes. But these chemicals have not previously been studied as auxiliary amendments for phytoextraction. There is a known study on the use of HEDP for phytoextraction of Cd, where HEDP has shown greater efficiency [44]. However, there are no studies on the effectiveness of using HEDP for polymetallic contaminants.
- The application of combined treatment with various functional corrections, which allows simultaneously stimulating the absorption of heavy metal ions, photosynthesis and biomass growth. The complex scheme should be based on a combination of treatments with a chelating agent, iron complexonate and hormonal supplements. Taking into account the pronounced manifestation of antagonism of metal ions during phytoextraction of multi-contaminated soils, the authors of this work put forward a hypothesis that the correction of iron deficiency can have a positive effect on (the overall efficiency of the process) stimulation of photosynthesis and improvement in the general physiological state of the latter. Furthermore, additional use of PGRs will also reduce the stress caused by high concentrations of pollutants and compensate for the negative effect of the latter on biomass growth.
9. It is also necessary to provide more informative data on the statistical processing of the obtained experimental data.
In section 2. Materials and Methods, an item has been added:
2.3. Statistical analysis
Data are presented as arithmetic mean and standard deviation or coefficient of variation, and the data were statistically compared between groups using Fisher’s test with Microsoft Office Excel 2007 software
10. A deeper analysis of the results is definitely needed in the context of a discussion of the other works on close and related topics. The latter should be presented in a wider spectrum.
The analysis of the results was revised and new works were added to the list of references, including on the use of clover for phytoremediation in Russia.
Reviewer 2 Report
This manuscript applied chelators and plant growth regulators to enhance the efficiency of metal phytoextraction by the tested plant. It is attractive and match the goals of sustainable and green technology. However, please improve points as below:
1. Many reviews for the background in the source of soil heavy metal and impact on plants were introduced, but the potential of phytoextraction for Trifolium repens has not been explained from the literature. Please re-organize these.
2. What is universal soil? Because the results of this experiment were controlled by the bioavailability of metal to plant in the pot tested soils that properties are very important in phytoextraction efficiency.
3. The findings: according to the concentrations of Cd, Cu, and Ni in the shoots, the plant did not show as a hyperaccumulator but the TF can increase over 1.0 or more in some treatments. Hence, the authors should highlight the novelty of this study for the enhancement of phytoextraction by these amendments.
Author Response
This manuscript applied chelators and plant growth regulators to enhance the efficiency of metal phytoextraction by the tested plant. It is attractive and match the goals of sustainable and green technology.
The authors are grateful to the reviewer for useful comments. We have highlighted in red the main corrections in the text of the manuscript.
However, please improve points as below:
1. Many reviews for the background in the source of soil heavy metal and impact on plants were introduced, but the potential of phytoextraction for Trifolium repens has not been explained from the literature. Please re-organize these.
The following paragraph has been added to the article:
‘The Trifolium repens L. (the white clover) seedlings. Clover is a very common wild-growing crop in the territory of the Russian Federation. In total, there are more than 70 growing species of clover within the country. Trifolium repens L., belonging to leguminous herbaceous crops, is kept in the herbage for 2-3 years. The root system of Trifolium repens L. is rod-shaped, with highly branching lateral processes. The bulk of the roots is located in the soil layer 40-50 cm Trifolium repens L creeping and its varieties are not demanding on soils, but they develop well on clay and loamy types, tolerate the proximity of groundwater better than other legumes (85-90 cm), has a high winter and frost resistance. Trifolium repens L. can accumulate both anions and cations effectively. Trifolium repens L. can compensate for soil enzyme activity loss caused by heavy metal [48].
Several authors used Trifolium repens L. as test plants in the study of phytoextraction of heavy metals in the soil and climatic conditions of Russia, where large areas are soddy podzolic loamy soils [49, 50]. High efficiency of clover in absorbing Ni, Cu, Zn was noted.’
2. What is universal soil? Because the results of this experiment were controlled by the bioavailability of metal to plant in the pot tested soils that properties are very important in phytoextraction efficiency.
We added in this texTable 1 with the main characteristics of the universal soil:
Table 1. The main characteristics of the universal soil “SELIGER-AGRO EXO” [51].
|
# |
Characteristic name |
Characteristic value |
|
1 |
Packaging volume |
60 litters |
|
2 |
Origin of soil |
Natural high-moor peat, neutralized with lime, with the addition of complex mineral fertilizer |
|
3 |
The type of plants for which this type of soil is applicable |
Ornamental, deciduous, herbaceous |
|
4 |
Mass fraction of water |
Up to 70% |
|
5 |
Acidity (pH KCl) |
5 - 6 |
|
6 |
Nitrogen content (NH4 + NO3) |
100-180 mg / l |
|
7 |
Phosphorus content (P2O5) |
135-255 mg / l |
|
8 |
Potassium content (K2O) |
115-215 mg / l |
|
9 |
Packaging weight |
18 kg |
3. The findings: according to the concentrations of Cd, Cu, and Ni in the shoots, the plant did not show as a hyperaccumulator but the TF can increase over 1.0 or more in some treatments. Hence, the authors should highlight the novelty of this study for the enhancement of phytoextraction by these amendments.
Thank you. We have added this information to the conclusion.
Round 2
Reviewer 1 Report
The concept of the manuscript was initially good in a number of ways but nevertheless required serious improvement.
It should be noted that the authors have done an excellent job and improved the manuscript.
At the same time, it is seen that there is still potential for optimization and improvement of work.
Therefore, if the authors put in a little more effort, the manuscript will make a significantly greater contribution to the area of research under consideration and will be cited more.
It is desirable to revise and improve this MS once again through the conceptual and system approach more - it is so for the title, abstract, the whole text, conclusions.
Author Response
The authors are grateful to the reviewer for supporting this work. The manuscript has been improved, especially in terms of arguments and discussions of results and the analysis of previous publications in this area. The corrections made are highlighted in red in the text.
Reviewer 2 Report
The manuscript has been revised to become complete, except for the very long conclusions. So, please shorten them.
Author Response
The authors are grateful to the referee for supporting this work. The conclusions have been shorted. The corrections made are highlighted in red in the text.
Round 3
Reviewer 1 Report
Definitely, this manuscript has the potential for the greater conceptualization of the material there as a whole.
Therefore the recommendation to the Authors is to improve it more.
Author Response
The authors express their deep gratitude to the reviewer for the appreciation of their efforts to improve this publication.
Following the recommendations of the reviewer, the authors checked the English using the MDPI service. The introduction has also been significantly improved. The main elements are highlighted in red to the text and the conclusion. And the rest of the text has been edited.